# Global Warming Drives Transitions in Suitable Habitats and Ecological Services of Rare *Tinospora* Miers Species in China

**Huayong Zhang [1,2,\*], Zhe Li [1], Hengchao Zou [1], Zhongyu Wang [1], Xinyu Zhu [3], Yihe Zhang [4] and Zhao Liu [2]**

[1] Research Center for Engineering Ecology and Nonlinear Science, North China Electric Power University, Beijing 102206, China; 120212232023@ncepu.edu.cn (Z.L.); zouhc@ncepu.edu.cn (H.Z.); zhy_wang@ncepu.edu.cn (Z.W.)

[2] Theoretical Ecology and Engineering Ecology Research Group, School of Life Sciences, Shandong University, Qingdao 250100, China

[3] Dalian Eco-Environmental Affairs Service Center, No. 58 Lianshan Street, Shahekou District, Dalian 116026, China; suzaneilbeck@gmail.com

[4] School of Engineering, RMIT University, P.O. Box 71, Bundoora, VIC 3083, Australia; henin.zhang@rmit.edu.au

[\*] Correspondence: zhanghuayong@sdu.edu.cn; Tel.: +86-010-61773936

**Abstract:** *Tinospora* Miers is considered a valuable medicinal herb that is suffering from severe habitat degradation due to climate change and human activities, but the variations in its suitable habitats and ecological service values remain unclear, especially in the context of accelerating global warming. In this study, we employed the MaxEnt model to estimate the suitable habitat changes and ecological service values of three rare *Tinospora* (*T. craveniana*, *T. yunnanensis*, and *T. sinensis*) species in China under four climate change scenarios (SSP126, SSP245, SSP370, and SSP585) from 2041 to 2100. The results show that the suitable habitats of *T. craveniana*, *T. yunnanensis*, and *T. sinensis* are mainly distributed in Sichuan, Yunnan, and Guangxi, respectively. Under the future climate scenarios, the suitable habitat of *T. craveniana* and *T. sinensis* is projected to expand toward the northeast and north, while that of *T. yunnanensis* will contract toward the northeast. The mean diurnal temperature range is the main environmental factor affecting *T. craveniana* and *T. yunnanensis*, while the annual mean temperature is a more important factor affecting *T. sinensis*. In the SSP245 scenario, *T. craveniana* and *T. yunnanensis* are expected to have the highest ecological service values from 2081 to 2100, while they will be relatively consistent in other climate scenarios and chronologies. The case of water protection accounts for the highest proportion of the total ecosystem service values, except for the economic value. This study provides a scientific reference for the diversity conservation of these rare species.

**Keywords:** medicinal herbs; climate change; MaxEnt model; Green GDP assessment





## 1. Introduction

The global ecological environment is facing immense pressure due to climate change and the rapid development of society [1–3]. Changes in vegetation patterns resulting from climate warming may displace species or lead to biodiversity loss [4,5], impacting the ecological services provided by plants to the environment [6–8]. Rare medicinal plants at risk of extinction have received significant attention due to their high ecological value and potential response to global change [9,10]. Thus, a thorough comprehension of vegetation habitat changes and ecological service values may provide a scientific basis for the preservation of the entire ecosystem and its biodiversity, contributing to the sustainable development of humans and the planet.

Understanding changes in the distribution and ecological services of threatened plants is crucial to conserving species distributions. Species distribution modeling (SDM) is a powerful tool for investigating vegetation distributions and their responses to environmental factors [11,12]. Among the various species distribution models (SDMs), the maximum

entropy model (MaxEnt 3.4.4) is notable for its high simulation accuracy compared to other models [13]. Additionally, it offers the advantages of stable software operation, low sample size requirements, and flexible variable processing [14,15]. As a result, it has been widely used to predict changes in the fitness zones of many rare plants [16–20]. This is due to the fact that the results closely match the distribution patterns of the species in their natural states [21–23]. At the same time, alterations in the habitat can affect the ecological services it provides [24]. The ecological assessment method based on Green Gross Domestic Product (GDP) assessment is the result of economic activity that takes into account the impact of natural resources and environmental factors, and it is considered more accurate and reliable than other methods, such as the direct market method and indirect market method, due to its comprehensiveness, objectivity, comparability, and operability [25]. It can comprehensively and accurately assess the value of ecological services, including atmospheric regulation, soil and water conservation, recreational value, and biodiversity [26]. Therefore, the combination of the two methods can be used to comprehensively measure and assess the future distribution dynamics and ecological value of rare vegetation. This serves as a methodological basis for the conservation and utilization of the resource and provides recommendations for the sustainable development of the resource.

*Tinospora* Miers, belonging to the family Menispermaceae, is a genus of valuable medicinal herbs that are able to provide fever-reducing and detoxifying effects [27], promote swallowing, and relieve pain, and they also have a certain role in water conservation and soil retention [28]. Unfortunately, most of these species are currently endangered to varying degrees, according to the Red List of China's Biodiversity—Higher Plants Volume (2020). This is due to slow growth rates, severe growing conditions, and extensive habitat fragmentation caused by both anthropogenic and natural disturbances [29], particularly affecting *Tinospora sagittata* var. *craveniana* (S. Y. Hu) H. S. Lo (*T. craveniana*), *Tinospora sagittata* var. *yunnanensis* (S. Y. Hu) H. S. Lo (*T. yunnanensis*), and *Tinospora sinensis*. (Lour.) Merr (*T. sinensis*). Previous studies on *Tinospora* have primarily focused on its basic efficacy and pharmacological effects. However, the distribution and changes in its suitability for coping with climate change remain unclear, and the value of its ecological services has not been accurately assessed [30–33]. Therefore, studying these aspects in depth will not only enhance our understanding of the ecological role of this plant but also generate new ideas and methods for ecological environmental protection and sustainable development.

In this study, we employed the MaxEnt model to anticipate the geographical distribution modifications of three rare *Tinospora* species (*T. craveniana*, *T. yunnanensis, and T. sinensis*) under four climate scenarios over three time periods, as well as their ecological service values using ecological valuation based on Green GDP accounting. This study aimed to investigate (1) the current suitable habitats of three species of *Tinospora*; (2) trends in the species under the future climate change scenarios and their ranges; (3) the key environmental factors influencing the changes in the *Tinospora* genus; and (4) the ecological service value of each of the three species under the different scenarios using Green GDP assessment. The results will guide the regional planning of *Tinospora* species in terms of their responsible cultivation and protection and provide a scientific basis and reliable references for decision-making to address their management under climate change.

## 2. Materials and Methods

### 2.1. Data Collection and Processing

The database for *Tinospora* comprises distribution point data, climatic variables, topographic variables, and soil variables corresponding to China. Chinese map data were selected from the official website of the Ministry of Natural Resources (http://www.mnr.gov.cn/ (accessed on 14 July 2023)) with the review number GS (2019) 1823. The geographical distributions of *T. craveniana*, *T. yunnanensis*, and *T. sinensis* in China were obtained by searching the records of specimens and the records of field surveys. Data were collected from the Global Biodiversity Information Network (http://www.gbif.org (accessed on 14 July 2023)), the China Virtual Herbarium (http://www.cvh.org.cn (accessed on 14 July

2023)), the NSII National Herbarium (http://www.nsii.org.cn/ (accessed on 15 July 2023)), and the China Nature Digital Herbarium (http://cfh.ac.cn (accessed on 15 July 2023)). The information was subjected to a screening process to remove records of artificial cultivation, duplicates, and incorrect sample points. This led to a combined total of 55 entries for *T. sinensis*, 18 for *T. craveniana*, and 15 for *T. yunnanensis* (Figure 1).

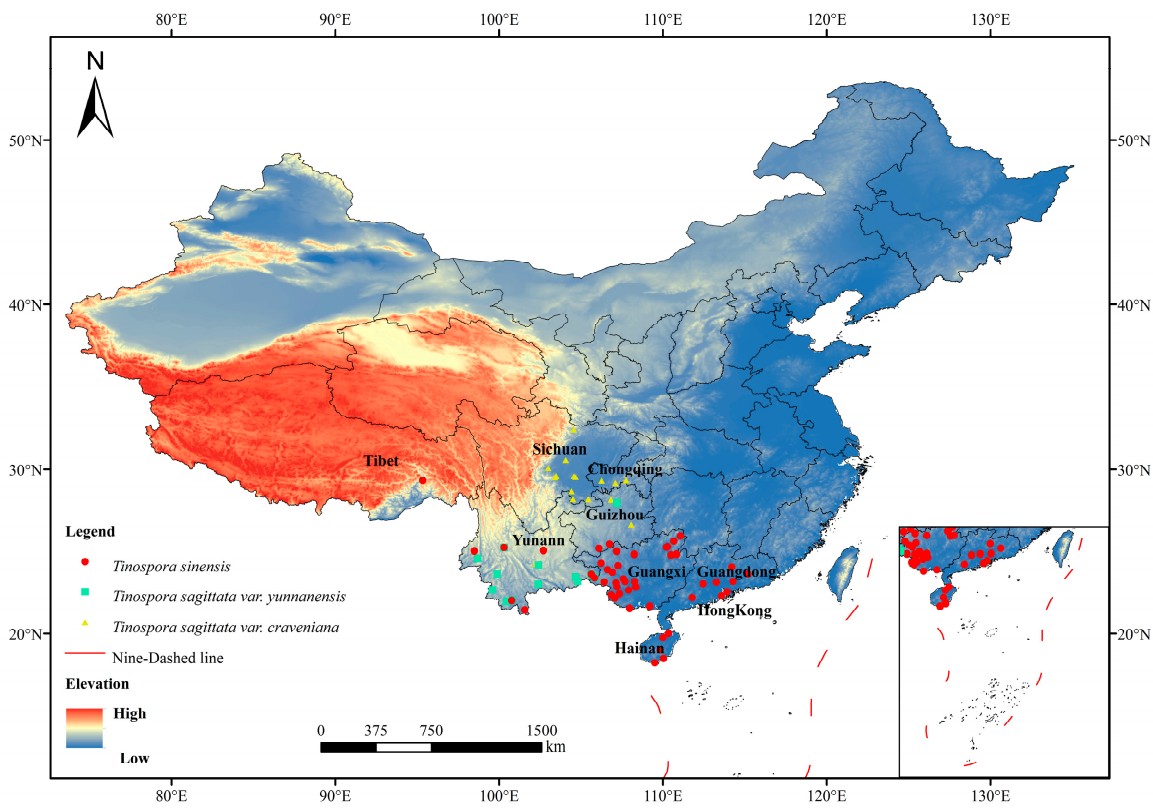

**Figure 1.** Distribution point data of *Tinospora* in China.

In this study, we initially selected 19 climatic variables, 3 topographic variables, and 11 soil variables. This study obtained climate data from the World Climate Database (https://worldclim.org/ (accessed on 18 July 2023)) for the contemporary period (1970–2000) and three future periods (2041–2060, 2061–2080, and 2081–2100). The data were collected at a spatial resolution of 2.5′ (5 km × 5 km). For the upcoming meteorological data, SSP126, SSP245, SSP370, and SSP585 are based on the four social sharing economy pathways projected in the BCC-CSM2-BR model from the CIMP6 (Coupled Model Intercomparison Project Phase 6) program. This was performed to generate four social sharing economy pathways comprising three different levels of greenhouse gas emissions: low (SSP126), medium (SSP245 and SSP370), and high (SSP585). Temperatures are expected to increase from low to high by 2.6 °C, 4.5 °C, 7.9 °C, and 8.5 °C [34]. Each scenario corresponds to 19 climate variables.

Topographic data were provided by worldclim 2.1 (https://worldclim.org/ (accessed on 18 July 2023)) at a 1 km spatial resolution. Soil data were obtained from the global SoilGrids version 2.0 database (https://soilgrids.org (accessed on 19 July 2023)) at 250 m spatial resolution. Then, the spatial resolution was standardized to 2.5′ (5 km × 5 km) using ArcGIS 10.8 software, which served as a basis for further research. In this study, topographic and soil factors were assumed to remain unchanged in the future time periods due to their low variability and minimal impact under global climate change scenarios.

To prevent overfitting in the species distribution model, the variables were screened using the variance inflation factor (VIF) [35] and the Pearson's correlation test to account for covariance between environmental factors. When the correlation coefficient between the

two factors exceeded 0.8, only one factor was selected [36]. Among the remaining factors, only those with a VIF of less than 10 were selected. This criterion was based on the study by Ranjitkar [37], who suggested that there was no multicollinearity problem when the VIF value between factors was less than 10. When screening the variables, we first used the results from the pre-runs in the maxent model to exclude the environmental variables with low contributions (percent contribution $\leq$ 1) [38]. Then, we subjected the environmental variables at each species site to Pearson's correlation test (Pearson's correlation coefficient r > 0.8, $p < 0.05$) [39]. When two variables were highly correlated, we selected the one with a relatively high contribution and high biological relevance [40–42]. The subsequent variance inflation factor (VIF) test was conducted using the criterion of retaining only environmental factors with values less than 10. This resulted in the selection of the following variables for the model (Table 1, Figure 2).

**Table 1.** The total examined environmental factors. Variables used for each species are also noted.

| Category | Variable | Description | Unit | Species [1] |
|---|---|---|---|---|
| Climate | Bio1 | Annual mean temperature | °C | b; c |
| | Bio2 | Mean diurnal temperature range | °C | a; b |
| | Bio3 | Isothermality | °C | c |
| | Bio4 | Temperature seasonality | °C | a |
| | Bio6 | Min temperature of coldest month | °C | a |
| | Bio7 | Temperature annual range | °C | a; b; c |
| | Bio8 | Mean temperature of wettest quarter | °C | |
| | Bio9 | Mean temperature of driest quarter | °C | |
| | Bio10 | Mean temperature of warmest quarter | °C | |
| | Bio11 | Mean temperature of coldest quarter | °C | b |
| | Bio12 | Annual precipitation | mm | c |
| | Bio13 | Precipitation of wettest month | mm | c |
| | Bio14 | Precipitation of driest month | mm | |
| | Bio15 | Precipitation seasonality | % | a |
| | Bio16 | Precipitation of wettest quarter | mm | c |
| | Bio17 | Precipitation of driest quarter | mm | c |
| | Bio18 | Precipitation of warmest quarter | mm | |
| | Bio19 | Precipitation of coldest quarter | mm | a; b |
| Topography | ASP | Aspect | degree | |
| | SLO | Slope | degree | a; c |
| | ELE | Elevation | m | |
| Soil | CEC | Cation exchange capacity of the soil | mmol/kg | a |
| | SOC | Soil organic carbon content in the fine earth fraction | dg/kg | b |
| | OCS | Organic carbon stocks | t/ha | a |
| | CLAY | Proportion of clay particles | g/kg | b |
| | SAND | Proportion of sand particles | g/kg | b |
| | pH.H$_2$O | Soil pH | pH × 10 | a; b; c |
| | BDOD | Bulk density of the fine earth fraction | cg/cm$^3$ | |
| | CFVO | Volumetric fraction of coarse fragments (>2 mm) | cm$^3$/dm$^3$ (vol‰) | |
| | NITROGEN | Total nitrogen (N) | cg/kg | |
| | SILT | Proportion of silt particles ($\geq$0.002 mm and $\leq$0.05 mm) in the fine earth fraction | g/kg | |
| | OCD | Organic carbon density | hg/dm$^3$ | |

[1] (a) *T. craveniana*; (b) *T. yunnanensis*; (c) *T. sinensis*.

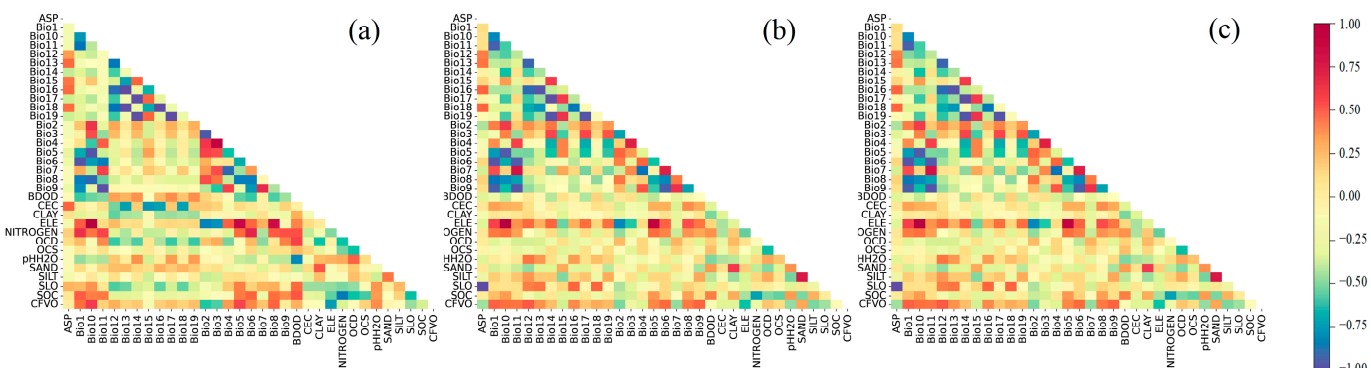

**Figure 2.** Results of the Pearson's correlation test for *Tinospora*. (**a**) *T. craveniana*; (**b**) *T. yunnanensis*; (**c**) *T. sinensis*.

### 2.2. Suitable Habitat Prediction and Migration Analysis

The *T. craveniana*, *T. yunnanensis*, and *T. sinensis* distribution points, along with the aforementioned environmental variables, were imported into MaxEnt 3.4.4 software. We chose a random selection of 75% of the sample data as the training dataset, while the remaining 25% were allocated to the test dataset [43]. To ensure the stability of the model, the model run was repeated ten times using the bootstrap method [44], and the weight of each environmental factor was measured through the jackknife technique. The findings were presented in the logistic format [18,45,46]. The outcomes were grouped into four classifications: highly suitable, moderately suitable, marginally suitable, and unsuitable habitats based on the natural discontinuity classification [47]. Subsequently, the area and percentage of each zone were calculated.

Receiver operating characteristic (ROC) curves were utilized to assess the precision of the MaxEnt model. The area enclosed by the ROC curve and its horizontal coordinate (AUC) are of constant size regardless of the critical value, thus rendering them useful for assessing the accuracy of prediction models [48–50]. An AUC value greater than 0.9 is considered to indicate the most effective simulation.

The time period from 2041 to 2100 was split into three intervals: 2041–2060, 2061–2080, and 2081–2100. Using the MaxEnt model, the possible geographic ranges of the three rare species of the genus *Tinospora* were simulated based on all four shared socioeconomic pathways: SSP126, SSP245, SSP370, and SSP585 under the future climate scenarios. The SDMtools plugin was utilized to determine the alterations in the habitats of the three rare species (expansion, contraction, and no change) in each time period and to determine the trend and direction of changes in the suitable habitats of the three species by comparing the different time periods under the same scenario [51]. The suitable habitats in the results were considered as a unit and condensed into a vector center of mass. The geometric center point's position denoted the overall spatial location of the appropriate region for *Tinospora*. Additionally, the displacement of the center of mass was applied to signify the species' overall spatial migration pattern in the suitable habitat [52].

### 2.3. Ecological Service Value Assessment

The three medical plants belonging to the genus *Tinospora* offer significant medicinal benefits as well as valuable contributions to biodiversity and ecological services. Furthermore, the ecological service value of *Tinospora* as a service varies depending on its future habitat distribution. Referring to the index system for assessing forest ecosystem service functions published by the State Forestry Administration [53] (LY/T1721-2008 forestry industry standard), this article focuses on measuring the value of herbal resources through Green GDP assessment. Additionally, the methodology below is based on the optimal habitat, as determined by the area component.

Atmospheric regulation value: Based on the photosynthesis equation, it is evident that for every gram of dry matter produced by a plant, 1.63 g of $CO_2$ is absorbed while 1.19 g of $O_2$ is released. The value of fixed $CO_2$ is calculated by taking an average from the carbon tax method and the silvicultural cost method. Similarly, the value of released $O_2$ is calculated by taking an average from the industrial oxygen method and the silvicultural cost method. This value is then used as the basis for determining the value of medicinal atmospheric regulation using the restoration cost method [54].

Water conservation value: The benefits of water conservation can be categorized into the advantages of medicinal flood control and the value provided by an increase in water resources. To determine the total amount of water conserved by medicinal plants, Equation (1) is utilized, while the total value is assessed using an alternative engineering method with a water storage cost of CNY 0.67 per tone [55,56].

$$Q = \rho \times H \times A \tag{1}$$

where $Q$ represents the quantity of water that medicinal plants contain, $\rho$ denotes the density of water, and H denotes the average water holding capacity of the 2 m soil layer, taking the median value of 890 mm. The distribution area of the genus *Tinospora*, which is calculated only within its highly suitable habitat, is represented by the variable A in this paper.

Soil conservation value: The soil conservation value is determined using the appropriate market value and alternative cost methods, taking into account the role of soil conservation and fertilizer preservation in the cultivation of medicinal plants. Therefore, the calculation follows Equation (2) [55,57].

$$E = \sum A \times C_i \times P \tag{2}$$

where the fertilizer retention value of the soil ($E$), the soil retention amount (A) represented by farmland in the middle and lower reaches of the Yangtze River, and the pure content of nitrogen (N), phosphorus (P), and potassium (K) in the soil ($C_i$) are defined, and the average price of chemical fertilizers in China (P) is CNY 2279 per tone.

Recreational value and biodiversity value: This study utilized a willingness to pay survey and the tourism expenditure method to objectively assess the recreational value of non-market goods and services. Following Wang et al.'s research on the tourism value of arable land resources, the recreational value of cultivated land for Chinese herbal medicine was quantified at CNY 225 per hectare [58]. The value of biodiversity was determined using direct market appraisal and conditional valuation methods. The resulting average value of maintaining biodiversity per unit area of agricultural ecosystems in China was quantified at CNY 628 per hectare [59].

## 3. Results

### 3.1. Contemporary Suitable Habitats

The MaxEnt results indicate that the suitable habitat for *T. craveniana* is mainly located in southeastern Sichuan and western Chongqing, accounting for about 8.89% of China's total land area. The main suitable habitat for *T. yunnanensis* is located in most parts of Yunnan, with a sporadic distribution in southern Tibet, accounting for about 7.88% of China's total land area. Meanwhile, *T. sinensis* is mainly distributed in Guangdong, Guangxi, and Hainan, covering approximately 8.36% of the land area (Figure 3, Table 2).

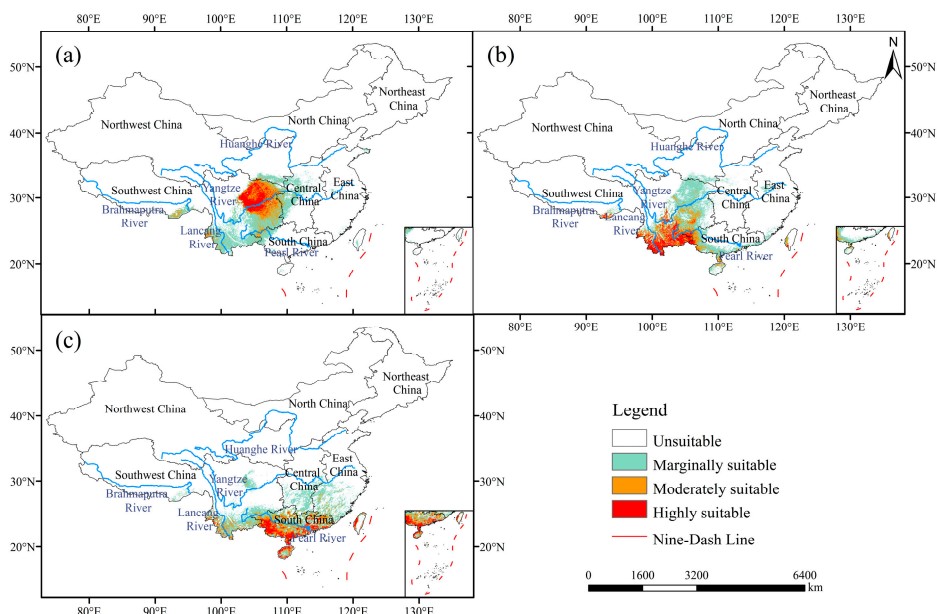

**Figure 3.** Suitable habitats for three species of *Tinospora* under the current climate scenario. (**a**) *T. craveniana*; (**b**) *T. yunnanensis*; (**c**) *T. sinensis*.

**Table 2.** Suitable habitat area ($\times 10^4$ km$^2$) and its ratio (%) for each species under the current climate scenarios.

| Species | Highly Suitable | | Moderately Suitable | | Marginally Suitable | | Total Suitable | | Unsuitable | |
|---|---|---|---|---|---|---|---|---|---|---|
| | Area | Ratio | Area | Ratio | Area | Ratio | Area | Ratio | Area | Ratio |
| *T. craveniana* | 11.90 | 1.24 | 19.78 | 2.06 | 53.66 | 5.59 | 85.34 | 8.89 | 874.66 | 91.11 |
| *T. yunnanensis* | 12.86 | 1.34 | 17.76 | 1.85 | 45.02 | 4.69 | 75.64 | 7.88 | 884.36 | 92.12 |
| *T. sinensis* | 10.18 | 1.06 | 18.43 | 1.92 | 51.65 | 5.38 | 80.26 | 8.36 | 879.74 | 91.64 |

The simulation results additionally showed that the spatial distributions of three species of *Tinospora* are influenced by various environmental factors, with temperature being the most significant contributor. The most dominant factor affecting *T. craveniana* (32.1%) and *T. yunnanensis* (24.0%) is the mean diurnal temperature range (Bio2), while the annual mean temperature (Bio1) is more important for *T. sinensis* (47.8%) (Table 3).

**Table 3.** Contributions of the environmental factors influencing each species.

| Category | Variable | Percent Contribution/% | | |
|---|---|---|---|---|
| | | *T. craveniana* | *T. yunnanensis* | *T. sinensis* |
| Climate | Bio1 | - | 6.3 | 47.8 |
| | Bio2 | 32.1 | 24 | - |
| | Bio3 | - | - | 4 |
| | Bio4 | 0.4 | - | - |
| | Bio6 | 31.5 | - | - |
| | Bio7 | 1.3 | 13.9 | 2 |
| | Bio11 | - | 15.6 | - |
| | Bio13 | - | - | 0.7 |
| | Bio14 | - | - | 4.3 |
| | Bio15 | 1.1 | - | - |
| | Bio16 | - | - | 15.9 |
| | Bio17 | | | 15 |
| | Bio19 | 9.1 | 6.3 | - |

**Table 3.** *Cont.*

| Category | Variable | Percent Contribution/% | | |
|---|---|---|---|---|
| | | *T. craveniana* | *T. yunnanensis* | *T. sinensis* |
| **Topography** | SLO | 0.5 | - | 3.8 |
| **Soil** | CEC | 16.6 | - | - |
| | SOC | - | 7.2 | - |
| | OCS | 0.6 | - | - |
| | CLAY | - | 8.6 | - |
| | SAND | - | 3.7 | - |
| | pH.H$_2$O | 3.6 | 5.3 | 6.5 |

*3.2. Dynamics of Suitable Habitats under Global Climate Change*

Under the future climate change scenarios, *T. craveniana* is expected to expand in a northeasterly direction. Under the SSP126 scenario, *T. craveniana* will contract by 3.14% of China's land area between 2041 and 2060, but it will also expand in a northeasterly direction by 3.41% of China's land area between 2061 and 2080 and by 3.56% between 2081 and 2100 (Figure 4a–c). Under the SSP245 climate scenario, *T. craveniana* is projected to expand, reaching a peak expansion area of 16.96% of China's total area between 2081 and 2100, mainly in the southern region where the species is widely distributed (Figure 4d–f). Under the SSP370 climate scenario, the distribution area of *T. craveniana* is expected to expand from southern Tibet to Liaoning in a northeasterly direction, and the proportion of the expanded Chinese land area is projected to rise from 17.74% to 43.71% (Figure 4g–i). Under the SSP585 climate scenario, the expansion trend of *T. craveniana* will gradually increase from the southwest to the northeast. The expansion area is expected to reach an extreme value of 6.40% of China's total area between 2061 and 2080 (Figures 4j–l and S1, Table S2).

Under the climate change scenarios, the potential habitat of *T. yunnanensis* is expected to shift northeast and contract. Under the SSP126 climate scenario, the territory suitable for *T. yunnanensis* is projected to decrease by up to 3.11% of China's land area between 2061 and 2080, resulting in the loss of suitable areas in Sichuan and Chongqing (Figure 5a–c). Under the SSP245 climate scenario, the suitable areas for *T. yunnanensis* are projected to decrease by 1.63% and 2.36% of China's territory between 2041 and 2080. However, there could be an expansion of the species' distribution in southwestern Xinjiang. The expansion area is expected to peak at 11.61% of China's total territory between 2081 and 2100 (Figure 5d–f). Under the SSP370 climate scenario, the distribution area of *T. yunnanensis* is projected to decrease by 2.59%, 2.88%, and 1.71% of China's land area between 2040 and 2100. However, there is a projected expansion of the suitable distribution area in southern Xinjiang from 2081 to 2100 (Figure 5g–i). Under the SSP585 climate scenario, the distribution area of *T. yunnanensis* is projected to decrease by 1.96%, 3.38%, and 2.38% of China's land area between 2040 and 2100. The trend of contraction for *T. yunnanensis* peaks between 2061 and 2080, regardless of the climate scenario (Figures 5j–l and S2, Table S2).

Under the four different climate scenarios, *T. sinensis* is expected to migrate northward and gradually increase in area. As the CO$_2$ concentration increases over time, the area of expansion will eventually increase to 8.23% of China's land area from 2081 to 2100 under the SSP585 scenario. By this time, *T. sinensis* will have spread over most of the southern part of the country Figures 6a–l and S3, Table S2).

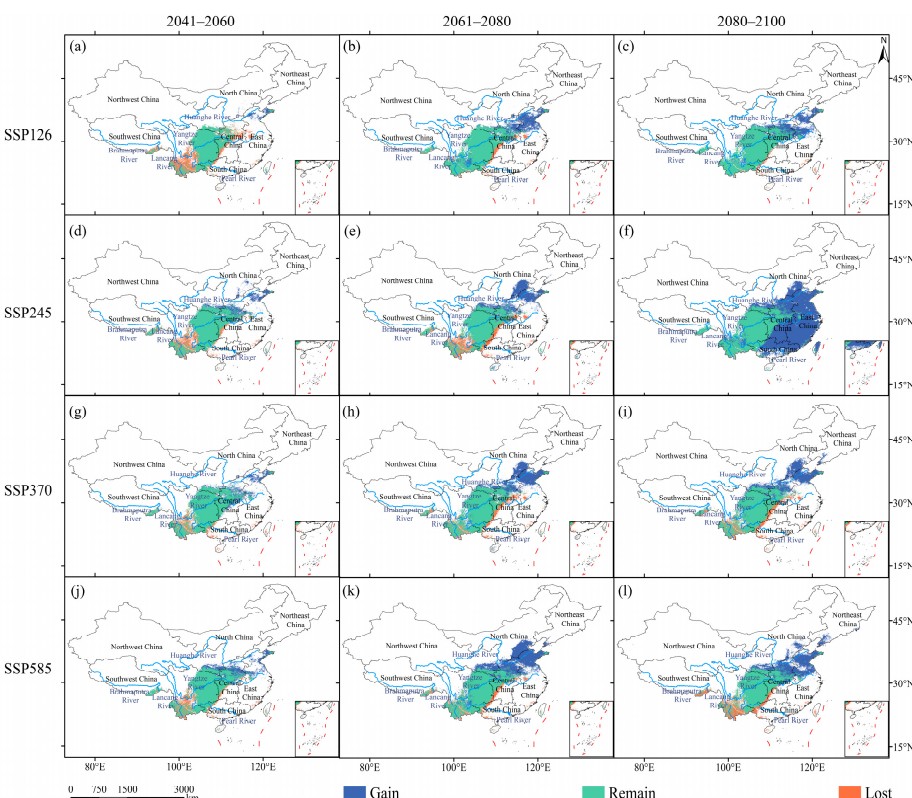

**Figure 4.** Changes in suitable habitats for *T. craveniana* under four climate scenarios during three time periods.

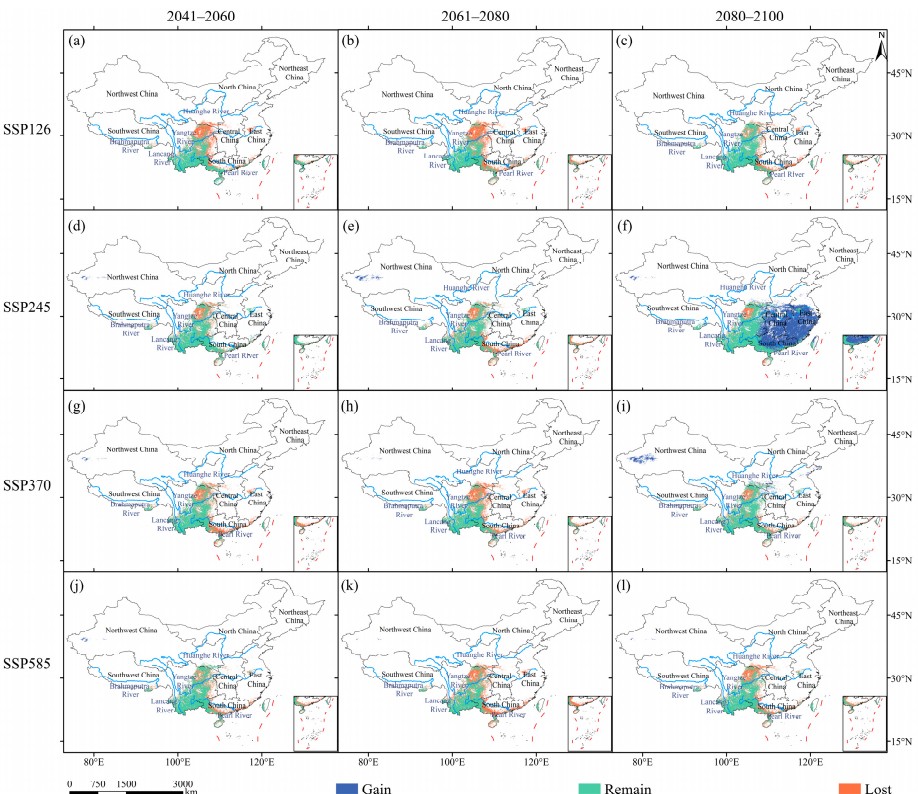

**Figure 5.** Changes in suitable habitats for *T. yunnanensis* under four climate scenarios during three time periods.

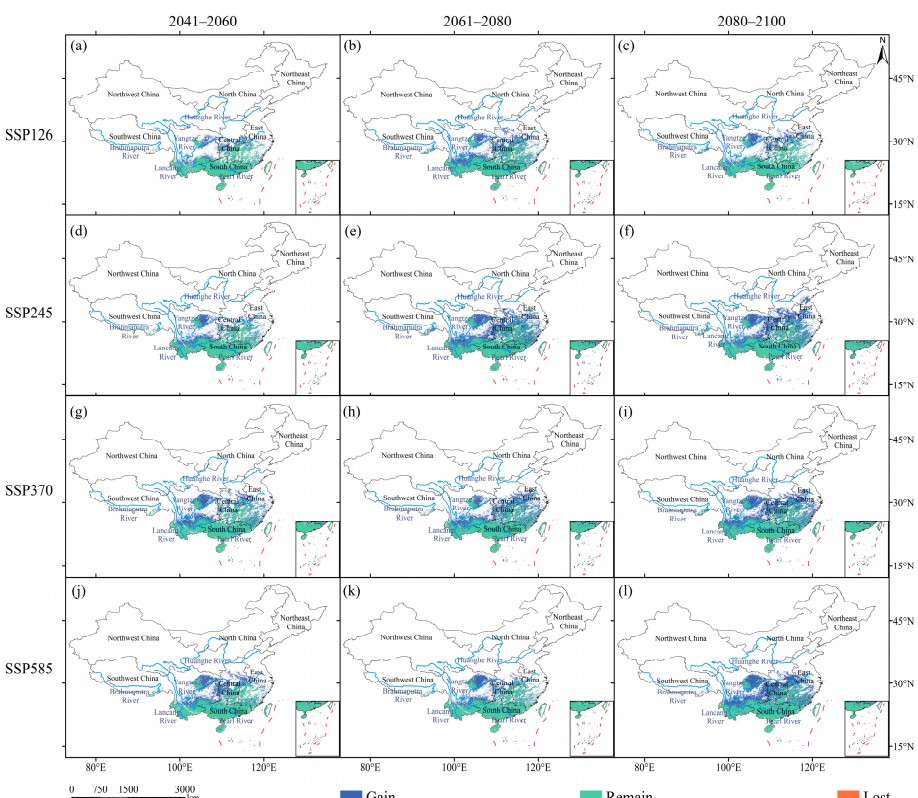

**Figure 6.** Changes in suitable habitats for *T. sinensis* under four climate scenarios during three time periods.

### 3.3. Ecological Service Value Assessments

The ecological services provided by *T. craveniana*, *T. yunnanensis*, and *T. sinensis* have various values, such as market economic value (71.38%), atmospheric maintenance value (2.48%), water conservation value (20.27%), soil conservation value (5.59%), recreational value (0.07%), and biodiversity value (0.21%). Among these values, the water conservation value is particularly noteworthy, alongside the rich medicinal market value (Table S3).

Under the SSP126 scenario, *T. craveniana* will show a contraction trend from 2041 to 2060 compared to the current state, resulting in an overall decrease in various ecological service values to 91% of the current values. However, from 2061 to 2100, these will gradually increase. For the remaining scenarios, the values of ecological services will increase to varying degrees. Between 2061 and 2080, the SSP245 scenario will result in the lowest growth rate of ecological services, which will be 1.01 times the current value. Specifically, between 2081 and 2100, under the SSP245 scenario, the value of ecological services will increase to 4.61 times the current value, reaching an extreme value (Figure 7, Table S4).

*T. yunnanensis* will exhibit an overall trend of contraction, except for the SSP245 scenario, where the ecological service value will increase to 2.29 times the current value during the period of 2081–2100. In the remaining scenarios, the ecosystem service values of *T. yunnanensis* will decrease to varying degrees. For example, in the three time periods of the SSP126 scenario, the values will decrease by 66.09%, 66.52%, and 72.41%. The most significant reduction will occur in the SSP585 scenario during the period of 2061–2080, where the value will only be 40.5% of the current ecosystem service value (Figure 8, Table S4).

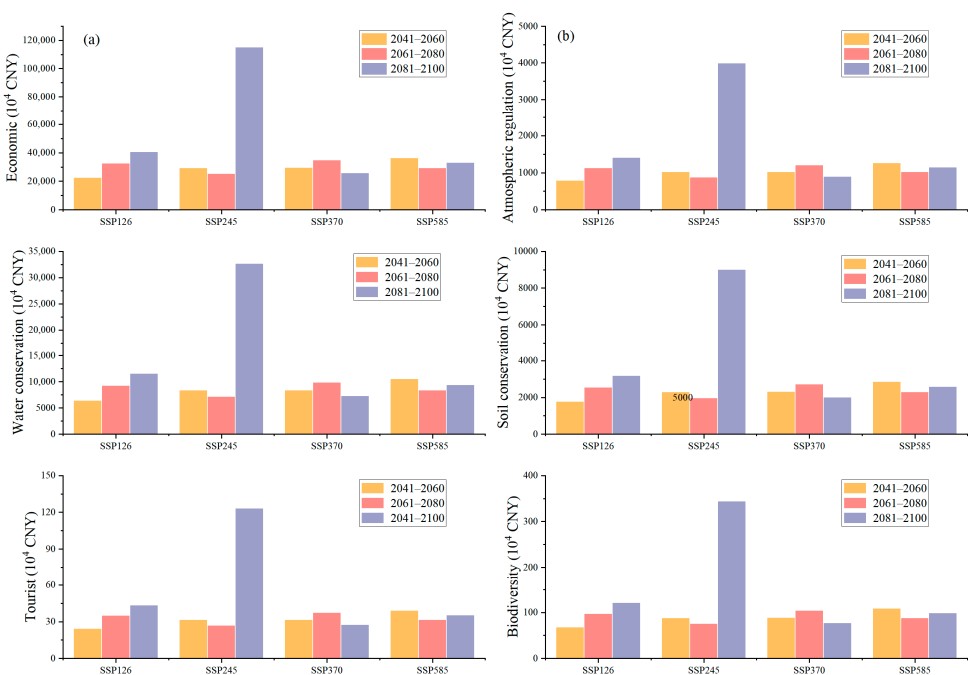

**Figure 7.** Detailed ecological service value of *T. craveniana*.

*T. sinensis* will increase in all scenarios considered. In particular, the SSP126 scenario shows the lowest rate of increase in *T. sinensis*, with each ecological service valued at 1.31 times the current value during the period 2041–2060. The greatest growth is projected to occur between 2081 and 2100 across all scenarios, with the SSP585 scenario showing an ecological value reaching 2.57 times its current value (Figure 9, Table S4).

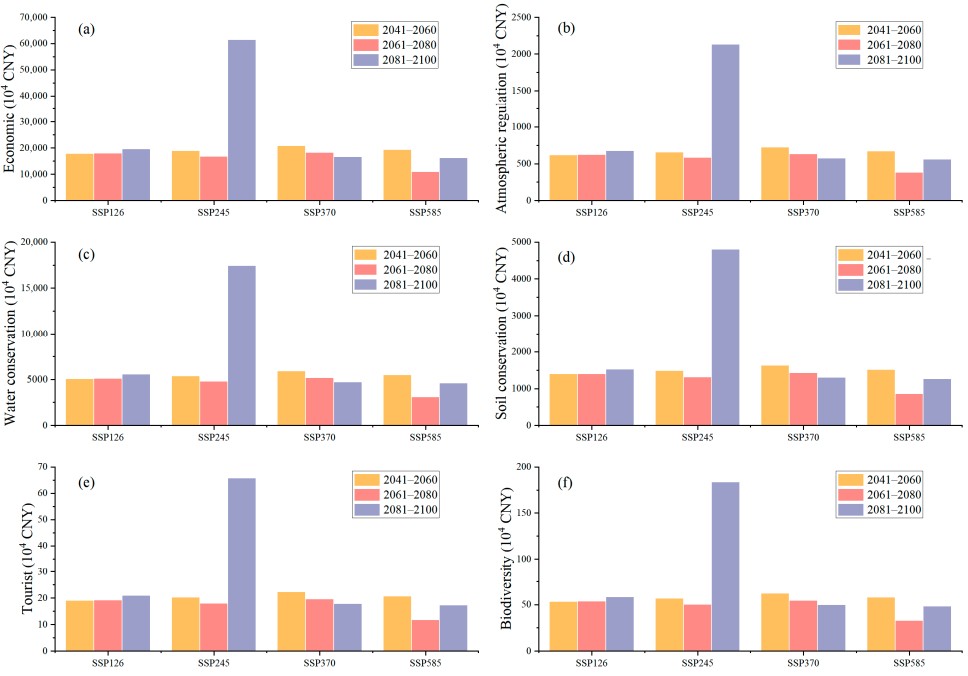

**Figure 8.** Detailed ecological service value of *T. yunnanensis*.

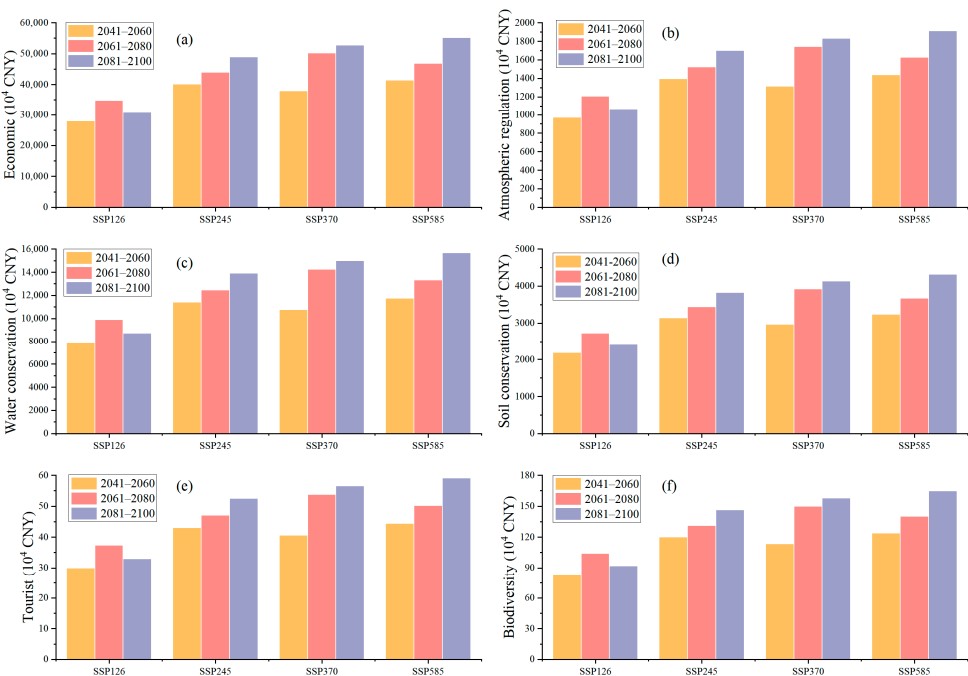

**Figure 9.** Detailed ecological service value of *T. sinensis*.

## 4. Discussion

This study predicts the current and future distributions of three *Tinospora* species under four climate scenarios for the period 2041–2100 using a MaxEnt model with distributional data and environmental factors. The model results indicate that the AUC values for the test sets of *T. craveniana*, *T. yunnanensis*, and *T. sinensis* are 0.988, 0.989, and 0.983, respectively. The AUC values for their corresponding training sets are 0.989, 0.989, and 0.988, all of which exceed 0.98. These results suggest that the model is accurate and reliable [60].

Research has indicated that climatic data and environmental variables can impact study results. In the present study, we modeled and predicted the habitable areas for three different rare species. We carried out several rounds of screening and testing of the data and variables, which allowed us to accurately predict the geographical distribution of each species. The results show that the suitable habitats of *T. craveniana*, *T. yunnanensis*, and *T. sinensis* will mainly be distributed in Sichuan, Yunnan, and Guangxi. These predictions align with the existing distribution locations of *Tinospora* plants and with the anticipated range of suitable habitat areas projected by the Global Medicinal Plants Geographic Information System (GMPGIS-I) in accordance with this discovery [61]. These data can serve as a theoretical framework for plant species' evolutionary processes and the preservation of rare *Tinospora* plants.

Trends in species alterations will fluctuate to some degree with the forthcoming climate warming and humidification situations [62]. *T. craveniana* is projected to expand its suitable habitat range in the northeast in different future scenarios. It may gain suitability in Hebei, Henan, and Shandong but may also lose some areas in Yunnan. This pattern is consistent with previous studies [63,64], indicating that the temperature has a beneficial effect on *T. craveniana* by accelerating the phenological process and prolonging the growing season. Additionally, the area of increase in suitable habitats is located on the boundary of suitable areas and is a sensitive area for species to respond to climate change [65,66]. *T. yunnanensis* struggles to adapt to the high warmth and humidity of its climate, leading to shrinkage. The suitable habitat will shift northeastward, resulting in unsuitable habitats in Sichuan, Chongqing, Guangxi, and Guangdong. However, according to some projections, *T. yunnanensis* may gain suitable habitats in parts of southwestern Xinjiang. This is due to the fact that, under the climate change scenarios, the southern border will have a climate more similar to that of the southern hemisphere and will be less susceptible to warming

and humidification than the northern border [67,68]. Perhaps due to its narrow ecological adaptation [64], the growth of *T. yunnanensis* as a liana may be negatively affected by the temperature [69]. There is a potential likelihood that, as *T. yunnanensis* is confined to the growth of the subtropical trees surrounding it, its suitable habitat may decrease proportionally with the decline in the population of those trees [70]. However, *T. sinensis* appears to be better adapted to climate warming. Based on various scenarios, there will be an increase in the area of suitable habitat to varying degrees. Additionally, the suitable habitat will migrate northward, indicating a tendency toward expansion. The expansion may reach provinces such as Yunnan, Fujian, Guangdong, and Guangxi. The overall trend of their migration aligns with the movement of flora and fauna to greater altitudes and latitudes in response to rising temperatures [50,71,72] Furthermore, the SSP245 scenario will result in a noteworthy expansion of the habitat area for *T. craveniana* and *T. yunnanensis* during the timeframe of 2081–2100. Hence, it is conjectured that the prevailing climate, temperature, and humidity conditions during this period will be exceedingly conducive to the thriving of both species.

Species survival depends on the surrounding environmental conditions, and temperature and humidity affect the physiological activities and biochemical processes of the entire plant group [73]. Under the different future climate scenarios, the three species of the genus *Tinospora* will show different adaptations to their environment. This study found that the main environmental variable affecting *T. craveniana* and *T. yunnanensis* is the mean diurnal temperature range (bio2), while the main environmental variable affecting *T. sinensis* is the annual mean temperature (bio1). The ability of temperature to influence their range is consistent with the fact that they are mainly distributed in the southern environments of China, along with two ecological factors, mean annual sunshine and mean annual relative humidity, that most affect the growth of the genus *Tinospora*, as found in previous studies [61]. In the process of global warming, the temperature gradually increases. *T. craveniana* and *T. sinensis* can still grow vigorously, reflecting the biological characteristics of their high temperature tolerance, but this also shows that the species of the genus *Tinospora* are hygrophilous, in accordance with their distribution in the undergrowth, along forest edges, in bamboo forests, or under the forests along the streams and valleys, and in other dark and humid places [74].

As three rare species of the genus *Tinospora*, they have not only a high medicinal market value but also an ecological service value, consisting of atmospheric regulation, water conservation, soil conservation, and the maintenance of biodiversity. Overall, their water retention capability constitutes a significant portion of the ecological services provided. As medicinal plants with a substantial amount of biomass and luxuriant foliage, these three species of this genus are recognized for their effective water storage capabilities and ability to prevent soil erosion [75]. *T. craveniana* has the widest suitable habitat area and demonstrates the largest suitable habitat under the SSP245 scenario from 2081 to 2100. Furthermore, of the three species across the various time periods and scenarios, this species is expected to exhibit the highest ecosystem service function values. These values are also related to the environment in which it grows and the species around it [76]. *T. yunnanensis* is experiencing a decline in population, and it is advisable to protect it in order to maintain its high ecological service function. Its optimum growth scenario is also the 2081–2100 period under the SSP245 scenario, and the scenario to be avoided is the 2061–2080 period under the SSP585 scenario. In contrast to prior research [77], our study indicates that in Yunnan Province, where it is largely located, the conservation value of water exceeds that of soil. We postulate that the habitat of *T. yunnanensis* may have declined in size, and, further, its restriction in water-stressed regions may have adversely impacted the trees on which it thrives, subsequently increasing the significance of water conservation [78,79]. However, *T. sinensis* exhibits strong adaptability to diverse scenarios of climate warming and humidification and demonstrates a gradual increase in its ecological service value over time. While its maximum value may not surpass that of *T. craveniana* in the preceding situations, it remains steady and progresses at a faster rate. Thus, it is suggested that more

attention should be paid to the conservation and cultivation of *T. sinensis* to achieve a high ecological service value.

As the three species within the genus *Tinospora* exhibit differing responses to climate change, their respective areas of required protection and recommended conservation measures also differ. *T. yunnanensis* is an endangered species that faces challenges adapting to future warming scenarios. To address this issue, we recommend the relocation of at-risk habitats and the establishment of new conservation areas within the highly suitable range. To achieve this, nature reserves or in situ conservation sites should be established in eastern Sichuan, Chongqing, and southern Yunnan. We emphasize the importance of minimizing human disturbance. Furthermore, it is feasible to cultivate *T. craveniana* and *T. sinensis* bladders locally in their highly suitable natural zones, namely Sichuan, Chongqing, Guangxi, and Guangdong.

## 5. Conclusions

*Tinospora* is a genus of medicinal herbs with high value that is currently endangered. However, the impact of climate change on its habitat distribution and ecological service value is unclear. To address this issue, we used the maximum entropy model to predict the distribution of *Tinospora* under future climate scenarios and assessed its ecological service value using Green GDP accounting, which can contribute to its conservation and long-term sustainable development.

This research revealed that the mean diurnal temperature range is the primary environmental aspect that influences the suitable habitats of *T. craveniana* and *T. yunnanensis*. Additionally, the main environmental determinant affecting the suitable habitat of *T. sinensis* is the annual mean temperature. *T. craveniana* and *T. yunnanensis* are primarily found in Sichuan and Yunnan in southern central China, while *T. sinensis* is mainly distributed in Guangxi and other regions of southern China.

For *T. craveniana*, all scenarios except the SSP126 scenario for the 2041–2060 period show northeastward expansion, with a potential gain in suitable areas in Shandong. *T. yunnanensis* shows northeastward contraction in all scenarios, except the SSP245 scenario for 2081–2100, with loss of suitable habitat in Sichuan and Chongqing. Under warmer and wetter scenarios, *T. sinensis* exhibits a northward expansion trend and may gain a significant amount of suitable habitat in Sichuan, Chongqing, and Anhui. Additionally, under the SSP245 scenario, between 2081 and 2100, *T. craveniana* and *T. yunnanensis* will reach their maximum area of suitable habitat.

Under the climate change scenarios, the ecosystem service values of *T. craveniana* exhibit an overall increasing trend, with the SSP126 scenario showing the fastest growth. Conversely, *T. yunnanensis* shows a decreasing trend, with the SSP585 scenario exhibiting the most severe decline between 2061 and 2080. Under the SSP245 scenario, *T. craveniana* and *T. yunnanensis* were found to have the highest ecological value from 2081 to 2100. From the SSP126 to the SSP585 scenario, the ecological service value of *T. sinensis* shows a general increasing trend over the three time periods. Notably, this study emphasizes the ecological service value of water conservation, which accounts for 20.27% of the total ecological service value, making it the highest value component. This suggests that the three species of *Tinospora* have a high value for water conservation, in addition to their economic value. This study provides a scientific reference for implementing conservation and management strategies.

**Supplementary Materials:** The following supporting information can be downloaded at: https://www.mdpi.com/article/10.3390/d16030181/s1, Table S1: Average AUC values of training and testing sets for each species; Table S2: Change of Suitable habitat area ($\times 10^4$ km$^2$) and its ratio (%) for each species; Table S3: Assessment of the ecological service value of genus *Tinospora* under present climate scenarios ($10^4$ CNY); Table S4: Future ecological service value of species of the Tinospora ($10^4$ CNY). Figure S1, Potential distribution habitats of *T. craveniana* under the future climate scenario; Figure S2, Potential distribution habitats of *T. yunnanensis* under the future climate scenario; Figure

S3, Potential distribution habitats of *T. sinensis* under the future climate scenario; Figure S4, Centroid migration of various species.

**Author Contributions:** Conceptualization, H.Z. (Huayong Zhang); methodology, H.Z. (Huayong Zhang); software, Z.L. (Zhe Li); validation, H.Z. (Huayong Zhang), H.Z. (Hengchao Zou), X.Z. and Y.Z.; formal analysis, Z.L. (Zhe Li); writing—original draft preparation, H.Z (Huayong Zhang)., Z.L. (Zhe Li) and H.Z. (Hengchao Zou); writing—review and editing, H.Z. (Huayong Zhang), Z.L. (Zhe Li), H.Z. (Hengchao Zou), Z.W., X.Z., Y.Z. and Z.L. (Zhao Liu); visualization, Z.L. (Zhe Li); supervision, H.Z. (Huayong Zhang) and H.Z. (Hengchao Zou); funding acquisition, H.Z. (Huayong Zhang). All authors have read and agreed to the published version of the manuscript.

**Funding:** This research was funded by National Water Pollution Control and Treatment Science and Technology Major Project (2017ZX07101) and the Discipline Construction Program of Huayong Zhang, Distinguished Professor of Shandong University, School of Life Sciences (61200082363001).

**Institutional Review Board Statement:** Not applicable.

**Data Availability Statement:** All links to input data are reported in the manuscript and all output. Data are available upon request to the authors.

**Conflicts of Interest:** The authors declare no conflicts of interest.

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
