# Peer review of "Global Warming Drives Transitions in Suitable Habitats and Ecological Services of Rare Tinospora Miers Species in China"

_diversity, doi:10.3390/d16030181_

Round 1

Reviewer 1 Report

Comments and Suggestions for Authors

 This study uses the MaxEnt model to assess future climate change distribution scenarios for three species in the genus Tinospora. These species have been studied for their medicinal value, but conservation issues have apparently not been addressed, so this study makes a potentially important contribution to the scientific basis for their management.

The main limitations of this study are the brief and missing descriptions of methods for some of the calculations. The abbreviations “SSP126, SSP245, SSP370, and SSP585” are not spelled out or described, so it is not apparent what future climate scenarios the authors are modeling. Furthermore, there is no information about how the ecological service values are calculated in Tables A3 and A4. There also needs to be some information about the green GDP accounting method, and how it is used to calculate the indices in the tables.

I have addressed these issues and several others in my comments below, and made some suggested edits to improve the English language. I suggest the authors request assistance from an English speaker to help them with edits, as additional language edits are needed beyond those listed below.

“Miers” should not be italicized. It is not part of the Latin name, but rather is the authority name (the scientist who named the genus).

34 “Global ecological environment….”

CHANGE TO

The global ecological environment….

40 “virtuous cycle”

I don’t know what this means, and cannot suggest alternate wording. I suggest this edit:

39 “a thorough comprehension of vegetation habitat changes and ecological service values may provide a scientific basis for preservation of the entire ecosystem and its biodiversity…..

54 GDP--spell out first time

54-56 Many readers will not be familiar with these different economic methods for ecological assessment. Please give a brief explanation. How is the green GDP accounting method used here?

63 “Tinospora Miers, belonging to the family Menispermaceae, is a valuable medicinal herb….

CHANGE TO

The genus Tinospora Miers, belonging to the family Menispermaceae, is a genus of valuable medicinal herbs… and they also have….

75-76 the species names should be in lower case (change throughout text)

81 “value of each genus”

I think you mean value of each of the three species.

84 “to address climate change”

CHANGE TO

to address their management under climate change

117 “Person” CHANGE TO Pierson

Table 1 “Soil oraganic carbon”. CHANGE TO organic

128 “We choose 75% of the sample data were selected randomly as the training dataset…”

CHANGE TO

We chose a random selection of 75% of the sample data as the training dataset…

140 “Where AUC value…”

CHANGE TO

An AUC value….

145 “SSP126, SSP245, SSP370, and SSP585”

Explain what these abbreviations mean. How did you decide what future climate scenarios to model?

178 “the mainly suitable habitat for T. Craveniana was mainly located in south-eastern Sichuan…”

CHANGE TO

the suitable habitat for T. craveniana was mainly located in south-eastern Sichuan…

180 “The mainly suitable habitat for T. Yunnanensis”

CHANGE TO

The main suitable habitat for T. yunnanensis….

192 Figure 2. Suitable habitats of three types of Tinospora Miers…

CHANGE TO

Figure 2. Suitable habitats of three species of Tinospora Miers under the current climate scenario.

198 “SSP126 scenario”

Again, you need to explain what these scenarios mean.

198 “contract 3.14% of China's land area…”

UNCLEAR. HERE IS A SUGGESTED EDIT:

contract within 3.14% of China's land area

199 “will expand [in] 3.41% and 3.56% of China's land area in a north-easterly direction between 2061 and 2100, respectively.”

I don’t understand this sentence. How can it expand both 3.41% and 3.56% during 2061-2100? There is a wording problem, but I don’t have a suggestion to correct this.

Figure 3, 4 legends have insufficient detail. Here is a suggested edit:  Changes in suitable habitats of ……. under four climate scenarios during three time periods.

Sec. 3.3 Ecological service values assessmen[t]s

Provide an explanation of how the ecological service values are calculated (Table A3, A4).

Also, do the plants need to have a certain density or percent cover to be able to provide these ecological services? Or I might ask this question another way: The authors write that the Tinospora species populations are depleted by disturbance. Do their populations need to recover before they can provide these ecological services?

282 “three salamander species”

Why are salamanders mentioned here? Is this an error? Please explain.

309 These liana species are dependent on tree understories for their growth. Are they associated with particular tree species, or are they generalists in selecting tree or forest communities? This would make a difference in their ability to expand habitat.

324 “main environmental variable affecting T. Craveniana and T. Yunnanensis was Mean diurnal temperature range (bio2), and the main environmental variable affecting T. sinensis was Annual mean temperature (bio1).”

The data reporting important environmental variables is not shown. These seem like important results to report in the Results section.

372 “Tinospora Miers is a medicinal herb…”

CHANGE TO

Tinospora Miers is a genus of medicinal herbs

Table A2

Correct spelling: Trand = Trend, Expasion = Expansion

Table A3, A4:  There is insufficient information on calculation of the indices for ecological service values. Please explain how these values are obtained.

Comments on the Quality of English Language

I suggested edits to improve the English language.  The authors might request assistance from an English speaker to help them with edits, as additional language edits are needed beyond those I suggested.

Author Response

Please see the attachment,Response to Reviewer 1 Comments

Reviewer 2 Report

Comments and Suggestions for Authors

Dear Authors,

I have thoroughly reviewed your manuscript titled “Global Warming Drives Transition in Suitable Habitat and Ecological Service of Rare Tinospora Miers Species in China” and appreciate the effort you've invested in your research. I find the study intriguing and potentially valuable for readers. However, before we can proceed with the publication, there are several key points that need clarification and revision. Please address the following concerns in your revised manuscript:

1-    Ensure there is no duplication of words from the title in the list of keywords.

2-    Include the author's name when introducing each species for the first time in the manuscript.

3-    Lines 26-27: Consider starting the sentence with 'In the case of water protection,' for better clarity.

4-    Lines 27-28: Could you please revise the punctuation in the following sentence? It appears there are two unnecessary dots at the end of the sentence that could be removed.

5-    Please refrain from using abbreviations in the keywords. For instance, instead of 'Green GDP assessment,' could you use the full term 'Green Gross Domestic Product assessment'? This will enhance clarity for readers.

6-    Lines 47-48: Add explicit references to supporting literature for claims, especially concerning the superior performance of the MaxEnt model. I recommend citing Damaneh et al. (2022) for further support. The reference is as follows: Damaneh, J. M., Ahmadi, J., Rahmanian, S., Sadeghi, S. M. M., Nasiri, V., & Borz, S. A. (2022). Prediction of wild pistachio ecological niche using machine learning models. Ecological Informatics, 72, 101907.

7-    Strengthen the connection between the reasons for endangerment and the study's objectives.

8-    Elaborate on the novelty of your study and specify the unique purposes not considered in previous research.

9-    Lines 91-92: Write the first letter of species and variety names in lowercase

10- Clarify the occurrence data or the number of presences for each species in each model.

11- Address the observed discrepancy regarding the assertion of no correlation between variables and the removal of certain variables.

12- Details on Variable Selection: Specify the names of topographic and soil variables, explain the criteria for removal, and clarify why certain variables were retained.

13- Variable Clarification in Table 1: Provide clarification on the variable labeled 'pH.H2O' in Table 1 and correct the term 'Oraganic carbom stocks' to 'Organic carbon stocks.'

 Please revise your manuscript accordingly. Your attention to these points will greatly improve the clarity, transparency, and overall quality of your work.

Author Response

Please see the attachment. Response to Reviewer 2 Comments

Round 2

Reviewer 1 Report

Comments and Suggestions for Authors

The authors have responded in detail to each reviewer comment, and the manuscript is much improved. I will recommend to the editor that it be accepted for publication. However, I have a few small recommended edits prior to publication:

65 “heat-clearing”

This is not a standard term used in English. Do you mean ‘fever-reducing’?

Response 15: “The climate data are from worldclim (https://worldclim.org) where the four future scenarios, ssp126, ssp245, ssp370 and ssp585, are based on the four social sharing economy pathways projected in CIMP6 (Coupled Model Intercomparison Project Phase 6). The four scenarios are low (ssp126), medium (ssp245, ssp370) and high (ssp585) in terms of greenhouse gas emissions. Each scenario corresponds to 19 climate variables.”

This is a very good explanation in response to Comment 15. However, the authors  need to add this information to the text, so readers will also gain a better understanding of the four climate scenarios (without requiring them to first read worldclim to understand your study). Also include in the text that the four scenarios model temperature increases of 2.6, 4.5, 7.9, and 8.5oC

359 “showe” CHANGE TO show

Author Response

Thank you for your feedback on this thesis. I have taken your suggestions into consideration and made revisions, which can be found at"Revised_manuscript"

Reviewer 2 Report

Comments and Suggestions for Authors

Dear Authors,

thank you for your effort to improve the manuscript, however, there are still a few points that require further attention and clarification in the manuscript.

- Only the first time you mention the genus or species, include the author's name. In your case, you wrote the author's name for the genus several times as (Tinospora Miers). Please edit it, and ensure that you include the author's name for these species: (T. craveniana, T. yunnanensis, and T. sinensis).

- In order to improve the abstract's effectiveness, it is suggested to enhance the first sentence by incorporating a brief introduction about Tinospora Miers.

 - I inquired about the methodology used to select which variable to remove in cases of correlated variables. Specifically, I would appreciate clarification on the criteria guiding the decision to retain certain variables while removing others in instances of correlation. Please provide insight into the reasoning behind these decisions in the next revision. Additionally, please begin by listing the names of all variables, followed by an explanation of which ones are correlated with each other and why you chose to retain specific variables. What criteria or priority influenced your decision-making process?

- Additionally, I had requested the author to incorporate the Pearson correlation results between variables into the manuscript. However, these considerations were not addressed in the current version. I kindly ask for your attention to these matters in the upcoming revision.

 -Least but not last, there are still issues related to punctuation and spaces between words. I kindly request your attention to address and improve these concerns for better readability and presentation.

Author Response

(The authors gave the same response as above.)
